# Carotid Atherosclerosis, Ultrasound and Lipoproteins

**DOI:** 10.3390/biomedicines9050521

**Published:** 2021-05-06

**Authors:** Arcangelo Iannuzzi, Paolo Rubba, Marco Gentile, Vania Mallardo, Ilenia Calcaterra, Alessandro Bresciani, Giuseppe Covetti, Gianluigi Cuomo, Pasquale Merone, Anna Di Lorenzo, Roberta Alfieri, Emilio Aliberti, Francesco Giallauria, Matteo Nicola Dario Di Minno, Gabriella Iannuzzo

**Affiliations:** 1Department of Medicine and Medical Specialties, A. Cardarelli Hospital, 80131 Naples, Italy; ale.bresciani@alice.it (A.B.); giucov@alice.it (G.C.); 2Department of Clinical Medicine, Surgery Federico II University, 80131 Naples, Italy; rubba@unina.it (P.R.); margenti@unina.it (M.G.); vania.mallardo@virgilio.it (V.M.); ileniacalcaterra@hotmail.it (I.C.); dario.diminno@hotmail.it (M.N.D.D.M.); gabriella.iannuzzo@unina.it (G.I.); 3Department of Translational Medical Sciences, Federico II University of Naples, 80131 Naples, Italy; gianluigi.cuomo95@gmail.com (G.C.); pasqualemerone.3@gmail.com (P.M.); dilorenzoanna2@gmail.com (A.D.L.); roberta.alfieri@libero.it (R.A.); francesco.giallauria@unina.it (F.G.); 4North Tees University Hospital, Stockton-on Tees TS19 8PE, UK; ealiberti@hotmail.co.uk

**Keywords:** lipids, atherosclerosis, carotid ultrasound

## Abstract

Carotid artery plaques are considered a measure of atherosclerosis and are associated with an increased risk of atherosclerotic cardiovascular disease, particularly ischemic strokes. Monitoring of patients with an elevated risk of stroke is critical in developing better prevention strategies. Non-invasive imaging allows us to directly see atherosclerosis in vessels and many features that are related to plaque vulnerability. A large body of evidence has demonstrated a strong correlation between some lipid parameters and carotid atherosclerosis. In this article, we review the relationship between lipids and atherosclerosis with a focus on carotid ultrasound, the most common method to estimate atherosclerotic load.

## 1. Introduction

This article is a review intended to illustrate the many ways that ultrasound is useful in the prevention of cardiovascular disease. We explain how the measurement of the carotid plaque burden is useful for risk stratification, for the assessment of the genetics and biology of atherosclerosis and for making decisions regarding treatment of patients. In addition, in this review we explore the relationship between lipids and atherosclerosis, the measurement of the intima-media thickness with carotid ultrasound and the evaluation of carotid plaques and carotid stiffness.

## 2. Relationship between Lipids and Atherosclerosis

Globally, stroke is the third largest cause of mortality and the largest cause of disability and therefore represents a very important public health problem [1]. Carotid artery atherosclerosis has a fundamental role in the occurrence of ischemic strokes, and it is estimated that 18–25% of thrombo-embolic strokes are caused by carotid pathology [2]. Accurately identifying those at high risk of stroke is essential for the implementation of better prevention strategies. In view of this, and in addition to the detection of conventional cardiovascular risk factors, non-invasive imaging allows us to directly see atherosclerosis in vessels and many characteristics that are associated with plaque vulnerability. There is a strong and well documented correlation between some lipid parameters and carotid atherosclerosis. The effect of lipids on subclinical atherosclerosis can be seen from childhood; children with heterozygous familial hypercholesterolemia have been shown to have an increased intima-media thickness in the carotid arteries [3,4]. Heterozygous familial hypercholesterolemia is an autosomal dominant genetic disease characterized by high levels of LDL-cholesterol (LDL-C) from birth, due to mutations in genes that code for the proteins involved in LDL metabolism. The most common mutations are those that code for the LDL receptor, whereas rarer mutations are for apoB and PCSK9. Familial hypercholesterolemia is associated with early and severe cardiovascular disease that is caused by increased amounts of LDL-C, for which pharmacological intervention is sometimes necessary from childhood [5]. Recently, Ference et al. [6] published an article on the impact of lipids on cardiovascular health, stressing the importance of the lipid load (which is derived from the exposure of an individual, over time, to high levels of atherogenic lipoprotein particles) on the progression of atherosclerosis. A recent study showed that cumulative exposure over time to high levels of cholesterol and triglycerides resulted in the appearance of new carotid plaques; the parameter that most influenced the appearance of new plaques was a high load (over time) of non-high-density lipoprotein cholesterol (non-HDL-C) [7]. Non-HDL-C includes low-density lipoprotein (LDL), intermediate-density lipoprotein (IDL) and very-low-density lipoprotein (VLDL). It is important to highlight that it is not only LDL but also lipoproteins rich in triglycerides (TRLs), which contain apo-B and have a diameter less than 70 µ, that pass through the endothelial membrane and tend to accumulate in the sub-endothelial space. Here they interact with proteoglycans in the extracellular matrix and induce a maladapted inflammatory response, dominated by macrophages and T lymphocytes, which promote the development of atherosclerotic lesions [8,9]. Therefore, it is not only LDL but also other lipoproteins that play a key role in the induction of vascular damage that can ultimately lead to a clinical event. The study of Iannuzzi et al. [10] on the association between subfractions of atherosclerotic lipoproteins and carotid atherosclerosis showed that in post-menopausal women cholesterol contained in VLDL and in IDL was significantly associated with carotid plaques, even after correction for other significant cardiovascular risk factors. In particular, the association between subclinical carotid atherosclerosis in women with higher levels of VLDL cholesterol was present even in patients without other cardiovascular disease and with normal or low levels of LDL cholesterol [11]. Among the least studied of the routinely investigated lipid parameters is small-dense LDL-C (sd-LDL-C), which is a group of LDLs with a density between 1044 g/mL and 1063 g/mL. Many studies suggest that sd-LDL-C is associated with a greater risk of myocardial infarction or stroke [12,13]. Sd-LDL-C is also positively correlated with the presence and progression of carotid atherosclerosis [14,15]. Lipoprotein (a) (Lp (a)) is a lipoprotein found in plasma that is composed of LDL covalently bound to apolipoprotein (a) with a disulphide bridge. Lp (a) is more atherogenic than LDL because the addition of apoprotein (a) promotes vascular inflammation and gives an antifibrinolytic effect with the inhibition of plasminogen [16,17]. Mendelian randomization studies suggest that Lp (a) could contribute causally to the onset of acute cardiovascular disease [18]. High levels of Lp (a) have also been found to be associated with an increased risk of ischemic heart disease, stroke, peripheral vascular disease and overall cardiovascular mortality [19]. Returning to the relationship between carotid atherosclerosis and lipids, it has been seen in perimenopausal women that there is an association between levels of Lp (a) and carotid atherosclerosis [20], an association that is similarly demonstrated in diabetic patients [21]. This can also be seen in patients undergoing endarterectomy, where increased levels of Lp (a) carried a higher risk of major adverse cardiovascular events [22]. Furthermore, in a recent study of diabetic patients it was demonstrated that there is an association between free fatty acids (FFA) and carotid plaques seen on ultrasound [23].

## 3. Carotid Ultrasound: Echography and IMT

Globally, the most common method of estimating atherosclerotic load has been carotid ultrasound. This is due to the ease of access to a superficial artery, the availability of high performance and reasonably low-cost ultrasound equipment and the presence of a large body of evidence (thousands of articles) that relates carotid artery pathology to cerebrovascular and cardiovascular events. The most frequent parameter studied in the extracranial carotid is the measurement of the intima-media thickness (IMT, Figure 1), as proposed by Pignoli et al. in 1986 [24]. Measurement of the IMT quickly established itself as a valid and reliable measure, with ad hoc protocols being developed for its use in numerous studies and drug trials, including use with lipid lowering drugs [25]. However, a quality control is needed to guarantee the reliability of results reported in both epidemiological and interventional studies. A large number of studies [26,27] have used the progression of the IMT as a surrogate marker for the initial stages of atherosclerosis [28,29], and a recent meta-analysis has suggested a positive association between the progression of the IMT and cardiovascular risk [30]. Recently, the value of carotid IMT for measuring subclinical atherosclerosis has been put in doubt [31], although Sirtori has subsequently defended it [32]. The ARIC study demonstrated in 2010 that the presence of carotid plaques in addition to IMT measurement was able to improve prognostic power in the prediction of coronary heart disease [33].

## 4. Carotid Plaques

The parameter that is most frequently considered in the evaluation of carotid plaques is the percentage of stenosis it results in, which is used for prognostic stratification and in determining therapeutic choices both in the United States and Europe [34,35]. Development of increasingly sophisticated ultrasound imaging techniques has allowed routine investigation of the characteristics of carotid plaques, something that was unimaginable with the first ultrasound devices. Given these premises, based on their intrinsic characteristics and regardless of the degree of stenosis that they result in, plaques are more likely to cause thrombo-embolic strokes. Over time this has led to the concept of vulnerable carotid plaques [36]. From a pathophysiological point of view, the transformation of a plaque from stable to unstable or vulnerable is characterized by a series of complex cellular and molecular mechanisms. We know that unstable and prone to rupture carotid plaques are characterized by a lipid-rich necrotic core and a thin fibrous cap. However, these aspects are better visualized on MRI scanning than by ultrasound [37,38]. The importance of carotid plaque characteristics as an independent risk factor for stroke has been recognized by the Vessel Wall Imaging Study Group of the American Society of Neuroradiology, who published guidelines focusing on current imaging techniques used in investigating the extracranial carotid artery and the importance of these modalities. This study generated three fundamental conclusions:The degree of stenosis is a weak indicator of the volume and extension of carotid plaquesSome intrinsic characteristics of carotid plaques seen on ultrasound are closely correlated with future coronary and cerebral eventsThese intrinsic characteristics of plaques significantly increase the risk of stroke, independent of the grade of stenosis [39].

The first criterion for characterizing a plaque at risk of causing stroke is the thickness and volume of the plaque. The larger the plaque, the greater the chance that it will cause an ischemic cerebral event. The Mannheim Consensus defined a carotid plaque as a “focal structure that encroaches into the arterial lumen of at least 0.5 mm or 50% of the surrounding IMT value or demonstrates a thickness >1.5 mm as measured from the media-adventitia interface to the intima-lumen interface” [40]. Other aspects that characterize carotid plaque instability are, as previously mentioned, a thin fibrous cap, the presence of a large lipid-rich necrotic core and the presence of hemorrhage in the plaque. There are few ultrasound (US) studies looking at thin fibrous caps, probably due to the poor sensitivity and reproducibility of ultrasound in their evaluation of this specific part of the plaque. The most well-known of these studies is that of Devuyst et al. [41], who in 2005 demonstrated good discrimination between symptomatic and asymptomatic carotid plaques following US measurement of the thickness of their fibrous caps. The presence of both a large lipid core and intra-plaque hemorrhage shows hypo-echogenicity of the plaque itself, and this has given rise to many studies looking at the echogenicity of carotid plaques.

Even toward the end of the 1980s, Gray-Weale et al. showed the importance of carotid plaque echogenicity in a comparative study between preoperative ultrasound appearance and carotid endarterectomy specimen pathology. They were able to demonstrate that plaques with a lower echogenicity on ultrasound preoperatively were more unstable and were associated with an increased frequency of hemorrhage and ulceration when plaques were examined histologically [42]. The same group successively proposed a system for grading carotid plaques based on the level of echogenicity; type 1 plaques had the lowest level of echogenicity, which increased until type 4, which had the highest level. A normal carotid artery was classified as type 5 [43]. Subsequent studies in the 1990s and early 2000s confirmed plaque echolucency was associated with an increased risk of stroke and TIA [44]. This has also been seen in prospective studies, which have shown an increased incidence of strokes on the ipsilateral side to echolucent plaques [45,46]. Furthermore, increased echolucency of plaques on carotid ultrasound is associated with an elevated risk of stroke in patients undergoing carotid stenting for stenosis secondary to plaques [47]. Recent data also show that carotid plaques with lower echogenicity more easily cause new cerebral ischemia following carotid endarterectomy [48]. In a meta-analysis involving 7557 asymptomatic patients who were followed for more than 3 years, it was found that the risk of ipsilateral stroke was twice as high in those with echolucent carotid plaques compared with hyperechoic plaques, irrespective of the severity of stenosis [49].

A method was proposed around this time for evaluating the echogenicity of carotid plaques with more objectivity and reproducibility than with the subjective judgement of the examiner alone: computerized grey scale median (GSM) was introduced. This method of evaluating plaques introduced an index of heterogenicity, which is represented by differences between the areas of highest and lowest echogenicity when using GSM measurement [50]. The development of GSM measurement led to another attempt to standardize ultrasound images, allowing greater reproducibility between different centers (using different scanners and sonographers). This was achieved by standardizing ultrasound images such that the GSM of blood in the carotid artery was between 0 and 5, and the GSM for the vessel adventitia was between 185 and 195 [51]. Grey scale median is defined as the median of the level of grey pixels in a plaque, and the percentage total of echolucent pixels is defined as the percentage of pixels with grey levels <40. The GSM value of an entire plaque is obtained from a histogram calculated by dedicated software.

The contents of a plaque are often heterogeneous, with fibrous and calcified components, a lipid or necrotic core and sometimes a hemorrhagic component. The proportions of these various components are variable across different plaques. In a study with a small number of patients, an attempt, using ultrasound and backscattering, was made to characterize these different “textures” of plaques by attributing a different color to each density (measured by GSM) present inside the plaque. In this study the color red was attributed to areas with low GSM values, yellow to intermittent values and green to higher GSM values [52]. For example, a large juxta-luminal lipid-rich necrotic core was associated with lower GSM values and therefore the color red predominated on the luminal surface of the plaque in these cases.

The question is what to do, as is often the case, when there are multiple plaques throughout the extracranial carotid artery? One approach, used by the Northern Manhattan Study, was to calculate the average GSM value across all plaques but to weight each plaque according to its size so that larger plaques had a greater influence on the result of that individual’s GSM. Subjects in this study were then divided into quintiles according to their weighted average GSM values [53].

## 5. Lipids and Low Carotid Plaque Echogenicity

Grønholdt et al. [54] have shown that plaques that have a lower density on ultrasound (those with a density closer to that of blood) are associated with elevated plasma levels of triglyceride-rich lipoproteins, both in fasting and non-fasting samples. In a subsequent study, the same group demonstrated that lower density (more echolucent) plaques contained increased lipid content on histological examination after carotid endarterectomy [55]. Researchers in the Tromsø study demonstrated that echolucent plaques were associated with low levels of HDL cholesterol, even in multivariate statistical analysis [56]. In elderly patients, the same association has been demonstrated between the echolucency of plaques and lower levels of HDL cholesterol [57]. Finally, in a study of a Chinese population, elevated levels of LDL-C and HDL-C were related to the presence of echolucent carotid plaques [58].

Are lipid-lowering therapies able to modify echogenicity of carotid plaques?

A meta-analysis has demonstrated that statins cause an increase in the echogenicity of carotid plaques [59]. Aggressive treatment with statins at high doses has been seen to increase the echogenicity of carotid plaques in patients with previous cerebrovascular events, making these plaques more stable and less dangerous [60,61]. Recently there has been an expansion in the use of new, and more potent, lipid lowering drugs capable of considerably reducing cholesterol levels thanks to the inhibition of proprotein convertase subtilisin/kexin type 9 (PCSK9). The physiological role of PCSK9 is to increase endosomal and lysosomal degradation of the hepatic LDL receptor (LDL-R), resulting in an increase in levels of LDL-C. Therefore, the inhibition of PCSK9 with monoclonal antibodies leads to increased expression and recycling of LDL-R on the cell surface, resulting in rapid clearance of LDL-C and a 50–60% reduction in cholesterolemia [62]. The addition of PCSK9 inhibitors to existing lipid-lowering therapy, and the achievement of very low levels of cholesterol, has brought about, in some case reports, morphological stabilization and regression of carotid plaques [63,64]. Overall, we can affirm that the presence of carotid plaques with a low echogenicity is an important prognostic factor in predicting future cerebrovascular and cardiovascular events, with an association between the carotid plaques and the plasma concentration of some lipid levels, particularly cholesterol contained in triglyceride-rich lipoproteins. This is useful for evaluating risk in dyslipidemic patients and can guide clinical management toward more aggressive pharmacological management for the reduction of lipids.

## 6. Carotid Neovascularization: CEUS and SMI

The formation of new, small blood vessels in the adventitia and in atherosclerotic plaques favors the development of intra-plaque hemorrhage, which plays a significant role in the transition from stable to unstable plaque and is therefore implicated in the evolution of clinically relevant complications [65]. Visualization of the adventitial vasa vasorum and intra-plaque neovascularization has consequently been recently considered as a possible new marker for atherosclerotic plaque instability [66,67]. Contrast-enhanced ultrasound (CEUS) has been shown to be a valid method of visualizing carotid neovascularization; this method involves the administration of intravenous contrast, which carries a very small associated risk. The accuracy of CEUS in the visualization of peri-adventitial and intra-plaque neovascularization has been confirmed by histological studies examining carotid plaques removed by endarterectomy [68]. The development of contrast agents for ultrasound, created from conjugated microbubbles that bind to specific ligands such as thrombi or neovascularization, has led to the creation of “molecular imaging”. These are innovative methods, still at the experimental stage, but microbubbles that bind specifically to key molecules in leukocyte trafficking such as P-selectin or vascular cell adhesion protein 1 (VCAM-1) have been used to document inflammatory phenomena, whereas microbubbles that bind to vascular endothelial growth factor (VEGF) can allow the study of neovascularization [69]. Another echographic method that allows the study of blood flow in micro-vessels without the use of contrast is “superb microvascular imaging” (SMI), developed by Canon Medical Systems Corporation, Otawara, Japan. Let us go into the details of these ultrasound techniques.

CEUS: This method is able to accurately identify many surrogate characteristics of carotid plaque vulnerability. The use of a contrast medium with specific microbubbles allows visualization of microulcerations and the plaque–lumen border to be optimized. It can also detail the presence of neovascularization in the plaque and in the adventitial vasa vasorum because these microbubbles are strictly intravascular [70]. In fact, the contrast medium is formed from microbubbles that contain a gaseous interior (normally sulphur hexafluoride (SF_6_)) encapsulated by a phospholipid capsule. Because this capsule is hydrophilic on the external surface and hydrophobic on the interior surface, it is sufficiently stable to allow changes to its shape and size; this allows it to oscillate, which is crucial to the formation of the CEUS signal. Moreover, these microbubbles have a diameter of 2.5 µ, which is a size that permits them to cross the internal vascular bed and allows them to reach even the smallest capillaries whilst being too large to cross the endothelium, which means that these bubbles remain intravascular. Another important characteristic of this contrast medium is that it is not solely excreted from the kidney because the phospholipid capsule is also metabolized by the liver, and the gas within is expelled via the respiratory system. This allows the use of CEUS in patients with renal impairment, which is frequently found in patients with vascular pathology. As previously described, CEUS investigation of the carotid artery gives clear definition of the border between the vascular endothelium and lumen, something that allows characteristics of the carotid plaque to be more accurately studied. It is therefore possible to verify if the surface of the plaque is smooth, irregular or ulcerated. The presence of ulceration of the plaque is a feature well known to be related to plaque vulnerability and carries a significant clinical and prognostic relevance for the possibility of future acute neurological events. An ultrasound investigation found that ulceration on an echolucent/hypodense carotid plaque increases risk of an ischemic cerebral event by 9-fold [71]. All those who deal with vascular US know that the diagnosis of an ulcerated plaque with ultrasound can be very difficult as there are many different definitions in the published literature, even amongst prestigious journals [72]. The criteria normally followed when discussing ulceration of carotid plaques on ultrasound were proposed by De Bray et al., who defined ulcers as greater than 2 mm depth with a well-defined back wall at its base on B-mode US and flow reversal present on color Doppler [73]. A definition of carotid plaque ulceration using CEUS is an interruption of the plaque–lumen border of at least 1.1 mm with filling of the interruption with microbubble contrast medium [74]. CEUS is, compared with the traditional method of echo-color Doppler, more accurate in the detection of ulceration, especially in terms of sensitivity [75]. Another important aspect of CEUS that outperforms echo-color Doppler is the possibility of studying neovascularization of the plaque, another characteristic that renders plaques unstable. Normally neovascularization/intra-plaque hemorrhage are classified subjectively based on 3 categories: 1. no visibility of microbubbles; 2. moderate visibility of microbubbles confined to the adventitial surface; 3. extensive presence of microbubbles inside the plaque. To allow reproducibility in different studies, software is being developed that is dedicated to quantifying the presence of micro-vessels present inside plaques automatically and objectively [76].

Superb Microvascular Imaging (SMI): We have seen that CEUS involves injection of intravenous contrast and that this is therefore something that would not be suitable for the routine study of carotid arteries but rather only in selected cases. As US techniques have developed, a more sensitive method (compared with color Doppler) for the study of low-speed flows has been developed called superb microvascular imaging (SMI). This method looks promising for the identification of neovascularization in plaques and therefore for the identification of unstable plaques, even if at present there are only a few studies that have documented its reliability in diagnostic or prognostic terms [77,78]. The characteristics of vulnerable plaque evaluated by different imaging techniques are shown in Table 1.

## 7. Lipids, CEUS and SMI

An Italian CEUS study that recruited patients with carotid plaques seemed to demonstrate that lower levels of LDL-C conferred protection against the neovascularization of plaques [79]. In a study conducted in China, the concentration of sd-LDL was positively associated with the presence of plaques that were classified as unstable with CEUS, and with logistical regression, sd-LDL was found to be an independent risk factor for unstable carotid plaques [80]. Moreover, the administration of atorvastatin in a group of 82 patients with carotid plaques showed a reduction in neovascularization in plaques studied with both CEUS and SMI [81]. In Table 2, we show the advantages and drawbacks of different ultrasound methods for characterizing carotid plaques.

## 8. Carotid Stiffness, Ultrasound

The mechanical properties of the carotid arteries can also be studied using US, and these functional changes are correlated with major cardiovascular risk factors [82]. Early diagnosis of cardiovascular changes in the initial stages of atherosclerosis is complicated, and further research into biopsy samples of young patients is needed, as was done in the PDAY study, which examined samples from young individuals who died of accidental causes [83,84]. Many studies have highlighted the importance of an increase in arterial stiffness as an early sign of vascular damage [85,86]. Multiple methods for measuring arterial stiffness have been implemented. Central stiffness of the aorta is usually measured using pulse wave velocity (PWV). Localized carotid stiffness can be measured with US or with echo-tracking [87]. Ultrasound enables us to measure many parameters of arterial elasticity (Table 3).

Some studies have shown a significant association between carotid stiffness and ischemic cerebral events [88,89].

## 9. Carotid Stiffness and Lipids

Carotid stiffness is elevated in children and women with metabolic syndrome [90,91]. Metabolic syndrome is meant as a cluster of cardiovascular risk factors (obesity, hyperglycemia with insulin resistance, hypertriglyceridemia, low levels of HDL-C and hypertension) whose simultaneous presence dramatically increases the incidence of cardiovascular disease and complications [92,93]. Increased carotid stiffness is observed even in subjects with familial hypercholesterolemia [94] and familial combined hyperlipidemia (FCHL). FCHL is the most common genetic hyperlipidemia and is characterized by the presence of hypertriglyceridemia and/or hypercholesterolemia in two or more members of the same family, with variable expression between individuals in a family or with multiple different phenotypes, and it gives an increased risk of early ischemic heart disease [95].

Interventional studies have shown that carotid stiffness improves in familial hypercholesterolemia following treatment with lipid lowering drugs [96].

## 10. Conclusions

In conclusion, carotid atherosclerosis can be studied with ultrasound using multiple methods, and it remains the first-line investigation for assessing the carotid artery. Advances in ultrasound are providing ever-improving images that now allow the characteristics of plaques to be evaluated, allowing us to assess the vulnerability of plaques and to risk stratify our patients accordingly. There is a documented correlation of the relationship between some lipid parameters and diseases of lipid metabolism, including the formation of plaques. We can also study the lipid load of an individual over time to further assess an individual’s risk. We can evaluate the progression and regression of carotid plaques and can monitor the effect of drugs or lifestyle that affect cardiovascular risk factors.

## Figures and Tables

**Figure 1 biomedicines-09-00521-f001:**
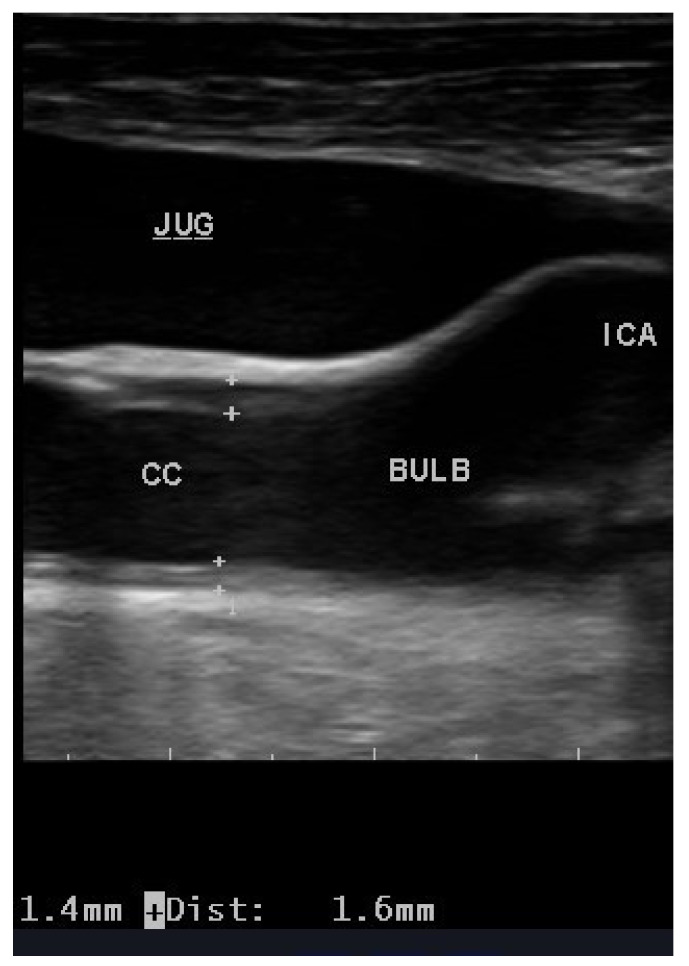
Increased IMT of the common carotid artery (1.4 mm far wall; 1.6 mm near wall). Legend: CC, common carotid; Bulb, carotid bulb; ICA, internal carotid artery; JUG, jugular vein.

**Table 1 biomedicines-09-00521-t001:** Vulnerable plaque characteristics with different imaging.

Histology	Echography	CEUS	T1-MRI	GD-MRI
Intraplaque hemorrhage	Echolucent	Echolucent	Hyperintense	Hyperintense
Lipid-rich necrotic core	Echolucent	Echolucent	Iso/Hyperintense	Hyperintense
Neovascularization	///////////	Enhance	///////////	Enhance
Inflammation	//////////	Enhance	//////////	Enhance
Ulceration	Irregularity	Irregularity	Irregularity	Irregularity

Legend: CEUS, contrast-enhanced ultrasound; T1-MRI, T1-weighted magnetic resonance imaging; GD-MRI, gadolinium magnetic resonance imaging; ///////////: not applicable.

**Table 2 biomedicines-09-00521-t002:** Advantages and drawbacks of different ultrasound methods for characterizing carotid plaques.

Method	How It Works	Advantages	Drawbacks	Final Output
Carotid US imaging, Gray Scale Median (GSM)	It is a computer-quantified index of echogenicity on ultrasound. Dedicated software calculates pixel brightness distribution based on the grayscale value of groups of pixels.	Simple and inexpensive method for calculating (with dedicated software) the echogenicity of a plaque.	Interframe variability in GSM during the cardiac cycle in US image sequences.	A quantitative value that allows the assessment of plaque echolucency and potential vulnerability.
Carotid Contrast Enhanced Ultrasound (CEUS)	Injection of microbubbles, which are strictly intravascular contrast agents. The signal produced by the microbubbles is acquired by the transducer and allows for real-time visualization of microbubbles for more than 4 minutes.	Compared with standard US imaging, CEUS allows a better visualization of the IMT, a more accurate outline of carotid plaques, neovascularization and precise detection of plaque ulceration.	CEUS is an invasive technique, and the cost of CEUS is relatively high. There is inconsistency between different visual grade classifications that compromises the accuracy of true positive intra-plaque neovascularization (IPN).	A visual quantification of intra-plaque neovascularization(IPN), which is an important feature of plaque vulnerability. A standardized visual grading system would be desirable for performing carotid CEUS.
Carotid Superb Microvascular Imaging (SMI)	The cornerstone of SMI is an algorithm for eliminating clutter that preserves low-flow signals. SMI is expected to detect very low-velocity blood flow and to allow visualization of micro-vessels, including carotid intra-plaque neovascularization (IPN).	It is a non-invasive technique. SMI allows the assessment of micro-vessels and the study ofthe vessel distribution in detail.	Only a few studies to date have documented its reliability in diagnostic or prognostic terms. Few US machines have this software.	Qualitative detection of neovascularization and of intra-plaque blood flow in carotid plaques.

**Table 3 biomedicines-09-00521-t003:** Measures of arterial elasticity by using ultrasound.

Parameter	Definition and Formula	Unit of Measure
Distensibility	The relative change in the area (or diameter) of the vascular lumen during systole for a given pressure range. DC = ΔA/A-ΔP or ΔD/D-ΔP	KPa^−1^
Compliance	Absolute change in the area (or diameter) of the vascular lumen during systole for a given pressure range. CC = ΔA/ΔP or ΔD/ΔP	m^2^KPa^−1^
Peterson’s elastic modulus	Inverse coefficient of distensibility. The change in pressure that induces a relative increase in the area (or diameter) of the vascular lumen. Ep = ΔPxA/ΔA or ΔPxD/ΔD	KPa
Young’s elastic modulus	Wall tension per cm of wall thickness for a 100% increase in diameter. YEM = ΔP-D/h x ΔD or (ΔP-Dd^2^)/(ΔD-2-IMT)	KPa
Stiffness (β)	Difference in pressure (transformed logarithmically) in relation to the relative change in arterial diameter. β = (Ln Ps- Ln Pd)/[(Ds-Dd)/Dd] or Ln (Ps/Pd)/(ΔD/Dd)	Unitless
Legend: Ps, systolic blood pressure; Sd, diastolic blood pressure; Ds, systolic (maximum) diameter; Dd, diastolic (minimum) diameter; ΔP, pulse pressure.		

## Data Availability

Not applicable.

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
