# Peer review of "Carotid Atherosclerosis, Ultrasound and Lipoproteins"

_biomedicines, 2021, doi:10.3390/biomedicines9050521_

Round 1

Reviewer 1 Report

The current review by Arcangelo Iannuzzi et al. summarizes association between carotid atherosclerosis and lipids and various noninvasive detection methods with a focus on carotid ultrasound. The review can be improved by following considerations.

1) The review should include an Introduction where the authors could briefly talk about carotid atherosclerosis, its association with lipids and various methods of detection and then move into the second section Relationship between lipids and atherosclerosis discussing all the details.

2) The authors should separate all their sections using numbers. 

3) The authors have repeated sentences from abstract in first paragraph and conclusion, they should either rephrase it or use similar studies to avoid repetition. 

4) There are no citations in few paragraphs, the review was not provided with line numbers so it’s difficult to point out those paragraphs. Authors should make sure they include proper citations.

5) Why is the table number reversed?

6) The review would be benefitted if authors could provide a table with all the variations of ultrasound methods and its characteristics that are available for measuring carotid plaque.

7)  Sentence formation, grammatical error and spell check required.

Author Response

1) The review should include an Introduction where the authors could briefly talk about carotid atherosclerosis, its association with lipids and various methods of detection and then move into the second section Relationship between lipids and atherosclerosis discussing all the details.

We have had a short introduction following reviewer’s suggestions

2) The authors should separate all their sections using numbers. 

Now all sections are separate using numbers (sections 1-8)

3) The authors have repeated sentences from abstract in first paragraph and conclusion, they should either rephrase it or use similar studies to avoid repetition

The abstract has been has been rewritten

4) There are no citations in few paragraphs, the review was not provided with line numbers so it’s difficult to point out those paragraphs. Authors should make sure they include proper citations.

All citations were checked

5) Why is the table number reversed?

Table number was amended

6) The review would be benefitted if authors could provide a table with all the variations of ultrasound methods and its characteristics that are available for measuring carotid plaque.

A new Table was added in the text following reviewer’s suggestion (see new Table 2)

7)  Sentence formation, grammatical error and spell check required.

A native English speaker reviewed the paper

Reviewer 2 Report

The review article is very well organized and written. The article comprehensively reviews recent literature on non-invasive imaging of carotid artery plaques with special focus on ultrasound. The authors thoroughly discuss the relationship between lipids, lipoproteins and carotid atherosclerosis. They highlight the necessity to reliably discriminate between stable and vulnerable plaques based on established guidelines. New strategies for contrast enhanced ultrasound like the usage of microbubbles are discussed.  In this respect, it would have been interesting to mention potential technical shortcomings. The conclusion could be more detailed to provide an outlook and to discuss how the correlation of metabolic and diagnostic parameter could be translated into reality.

The paper is very interesting to read and provides a very good overview on existing and emerging approaches.  I would recommend to publish the manuscript in its present form.

Author Response

We thank the referee for the excellent judgment on the work

Round 2

Reviewer 1 Report

The manuscript is significantly improved and looks good.